

# Re-design of an upwind rotor for a downwind configuration: design changes and cost evaluation

Gesine Wanke[1], Leonardo Bergami[1], Frederik Zahle[2], and David Robert Verelst[2]

[1]Blade Science Center Suzlon, Vejle Denmark
[2]DTU Wind Energy, Technical University of Denmark, Roskilde, Denmark
**Correspondence:** Gesine Wanke (gesine.wanke@suzlon.com)

**Abstract.** Within this work, an existing model of a Suzlon S111 2.1MW turbine is used to estimate potential cost savings when the conventional upwind rotor concept is changed into a downwind rotor concept. A design framework is used to get realistic design updates for the upwind configuration as well as two design updates for the downwind configuration, including a pure material cost-out on the rotor blades and a new planform design. A full design load basis according to the standard has been

used to evaluate the impact of the redesigns on the loads. A detailed cost model with load scaling is used to estimate the impact of the design changes on the turbine costs and the cost of energy. It is shown that generally lower blade mass can be achieved with the downwind configurations of up to 5% less than the upwind redesign. Compared to an upwind baseline, the upwind redesign shows an estimated cost of energy reduction of 2.3% where the downwind designs achieve a maximum reduction of 1.3%.

# 1 Introduction

Historically the first wind turbines were dominantly downwind turbines, where the rotor was placed behind the tower, seen from the incoming wind. This turbine configuration was considered safer than the alternative upwind configuration with the rotor in front of the tower since the rotor blades would bend away from the tower under turbine operation. Early research, mainly

by NASA and associated partners compared the downwind rotor configuration with the upwind configuration. Glasgow et al. (1981) showed that the wake behind the tower caused significantly higher cyclic flapwise blade root loads in the downwind configuration compared to the loads of the upwind rotor configuration. However, neither for the flapwise mean blade root loads or for the edgewise blade root loads differences between the two configurations could be shown.

Many residents living near early downwind wind turbines reported high noise levels and especially the high unsteadiness, a

"thumping" sound was reported as a nuisance (Metzger and Klatte (1981)). The rotor blades passing through the tower wake caused high low-frequency noise and amplitude modulation. Measurements in an anechoic wind tunnel by Greene (1981) demonstrated that downwind rotors on a lattice tower radiated more noise than downwind rotors on a tubular tower due to the



narrower and deeper tower wake. Upwind rotor configurations, on the other hand, were found to be significantly less noisy. Upwind rotor configurations therefore dominated industrial applications as well as the focus in research efforts during the 1990s and 2000s.

Cost driven industrial designs prefer larger rotor areas to capture more energy. The rotor blades for modern sized upwind wind turbines are designed under a constraint of maximum blade tip deflection to avoid a collision of the blades with the tower. Since the blade tip deflection constraint could be eliminated in modern sized wind turbines, the downwind configuration is currently coming into research focus again, especially for future even larger rotors.


Advances in wind turbine noise mitigation techniques since the 1980s as well as airfoil design could overcome the previously reported noise issues, to an acceptable level. Reiso and Muskulus (2013) successfully eliminated the tower shadow effect on the fatigue loads by using a fairing. While the fairing is a rather costly device to implement the study further showed the potential that fatigue loads can be significantly reduced by a reduced flapwise stiffness, alleviating loads by blade deformation.

Ning and Petch (2016) used an optimization framework to compare the levelized cost of energy of land-based upwind and


downwind turbines. The study included turbines of different wind classes, rated power and rotor diameter. Modest cost savings could be achieved for the downwind configuration compared to the upwind configuration for wind turbines of wind class III. Blade mass savings had to offset the higher tower cost originating from the increased tower bottom bending moment as the gravity overhanging moment of the rotor-nacelle assembly coincides with the moment from the thrust force.

In a system-level design study for large rotors Zalkind et al. (2019) showed that coned downwind rotors significantly reduce


peak blade loads during operation but have a lower annual energy production compared to a coned upwind configuration of the same size. While the group predicts larger main bearing peak loads for the downwind configuration related to blade length, mass, and cone angle they suggest that the increased tower loads observed by other groups could be overcome by properly balancing the nacelle on the tower.

A reduced edgewise damping for a downwind configuration compared to an upwind configuration was identified by Wanke


et al. (2019a), leading to significantly higher edgewise loads in the downwind configuration than in the upwind configuration. In a following study on a 2.1MW turbine Wanke et al. (2019b) showed that large downwind cone angles could reduce the edgewise damping further, as the out-of-plane contribution of the edgewise mode shapes is decreased. A significantly reduced tower torsional stiffness, on the other hand, e.g. a lattice tower configuration would benefit the downwind configuration.

Aligning the blades with the loading direction of aerodynamic forces, gravity, and centrifugal force is an opportunity of the


downwind configuration to significantly reduce flapwise bending loads, loading the blade in axial tension instead. Such a load distribution is achieved by adjusting the cone angle and blade prebend. These downwind rotors with so-called "load alignment" have been suggested as an option to reduce blade mass significantly, utilizing large cone angles and downwind prebend by Loth et al. (2017) for a 13.2MW wind turbine. The study also indicated mass savings compared to the conventional upwind rotor when the blade length is increased to compensate for energy production losses.


Bortolotti et al. (2019) used an optimization framework to compare the cost efficiency of an upwind configuration with a downwind configuration and a downwind configuration with "load alignment". The analysis for a 10MW turbine showed difficulties to reach a more cost-efficient design for the downwind configurations than the conventional upwind configuration,





due to other component costs.

Often proposed are downwind configurations with a passive wind direction alignment. Such yaw systems could be cost-efficient
as they simplify the turbine control, and reduce operation and maintenance costs, as they could purely be used for cable
unwinding. However, Wanke et al. (2019c) showed on an example of a 2.1MW turbine with a tilted rotor that such systems
align passively at high yaw angles for high wind speeds resulting into significant power loss. The study concluded that tilt
angle, cone angle and blade stiffness would need to be specifically designed for a free yawing downwind configuration. This
would expose additional constraints on a downwind turbine design, while the benefit in terms of a cost-efficient turbine is
questioned.

The cost-efficient design of wind turbines has been approached to a growing extent by the use of optimization frameworks.
Over the years it has been questioned that rotors designed for the maximum efficiency result in the most cost-efficient turbine
designs. Optimizing a conventional 10MW upwind turbines for the lowest cost of energy (COE) Bottasso et al. (2016) showed,
that designing the rotor for minimum cost instead of maximum annual energy production (AEP) results into rotors with larger
chord, higher thickness and lower AEP. Higher absolute thickness could utilize higher stiffness with less material resulting in
the lower cost compensating the AEP -loss from the less efficient, thicker airfoils.

Lower rotor loads could potentially result in the possibility to increase the rotor length and therefore increase the overall power
capture. This could be a more cost-efficient rotor than a traditional design approach, also for upwind turbines. Bottasso et al.
(2015) tried therefore to design a low induction rotor for a 10MW wind turbine with an optimization framework where the
blade shape was designed with the common aerodynamic parameters, such as chord, twist and airfoil thickness. Their work
showed that maximum AEP solutions might be achievable with low induction rotors, but the minimum cost solutions might be
different from the maximum AEP solutions. It was seen to be very dependent on the cost model if the higher AEP could pay
for the increased rotor diameter.

Loenbaek et al. (2019) investigated design trends by an optimization of power capture based on 1D momentum theory. Their
work indicated that the maximum power capture is achievable by a larger rotor diameter and operation at lower cp than
maximum. For a conventional upwind turbine, this is achieved by so-called thrust clipping or peak-shaving. The peak-shaving
is a control feature that reduces extreme flapwise loads as well as it increases the minimum blade tip to tower clearance in the
upwind configuration while sacrificing AEP.

This paper evaluates the specific example of the Suzlon S111 2.1MW turbine the potential of a downwind turbine configuration
compared to the original upwind turbine configuration regarding mass and cost reduction. It is shown that a 5% lower rotor
mass can be achieved in the downwind configuration than in the upwind configuration. Despite higher tower and foundation
costs the turbine specific cost model indicates lower capital expenditures (CAPEX) for the downwind configuration than the
upwind configuration. Due to the difference in AEP, the upwind configuration is the most cost efficient configuration with a
1.0% lower COE.





## 2   Methods

This work aims to compare an upwind configuration of an existing turbine with a downwind configuration, from a cost and mass perspective. The chosen example turbine is the Suzlon S111 2.1MW turbine, a commercial upwind turbine. The turbine is designed for wind class IIIA, with glass fiber blades, a rotor diameter of 112m on a 90m tubular tower. The turbine is pitch regulated with a variable speed generator. The shaft is tilted, the rotor is coned and the blades are prebend. All three geometrical parameters increase the blade-tip to tower clearance in the upwind configuration.

For this turbine, a new baseline rotor blade is defined, inspired by the commercial blade, which is adapted to the framework. For the baseline rotor, an upwind turbine configuration is generated, called S111uw. Additionally, a downwind baseline turbine configuration is defined with the baseline rotor called S111dw. The downwind configuration utilizes the same cone and tilt angle, both increasing blade tip to tower distance. Since the blade prebend of the rotor is towards the blade pressure side the prebend decreases the blade tip to tower distance in the downwind configuration. Three rotor redesigns are made. For the upwind configuration, a blade planform and internal structural redesign is made. The design is called S111uw PF. For the downwind configuration, two scenarios are regarded. Firstly, a pure blade material reduction is performed, called S111dw STR. This corresponds to a configuration change of an existing upwind configuration into a downwind configuration while keeping the blade molds but saving blade material. Secondly, a blade planform and structure redesign in the same manner as for the upwind redesign called S111dw PF. Table 1 shows a summary of the design configurations regarded as well as a name indicator used throughout the study.

The rotor design procedure uses a low fidelity optimization tool to create a blade planform and stiffness distribution. The

**Table 1.** Turbine configurations regarded in design and cost estimation

| Name | configuration | planform | structure |
|------------|---------------|-----------|-----------|
| S111uw | upwind | baseline | baseline |
| S111uw PF | upwind | optimized | optimized |
| S111dw | downwind | baseline | baseline |
| S111dw STR | downwind | baseline | optimized |
| S111dw PF | downwind | optimized | optimized |

planform and stiffness distribution are afterwards matched within the HAWTOpt2 framework Zahle et al. (2015, 2016) to create a full HAWC2 (Madsen et al. (2019)) set-up for aeroelastic load calculations. For all designs, a full design load basis (DLB) is calculated. The loads are used to calculate a failure index of the blades, to evaluate if the redesigns are acceptable. From the tower loads, the required tower material is calculated. Finally, the costs of all five designs are calculated with a load and mass scaling cost model. This design procedure is conceptually outlined in Figure 1.



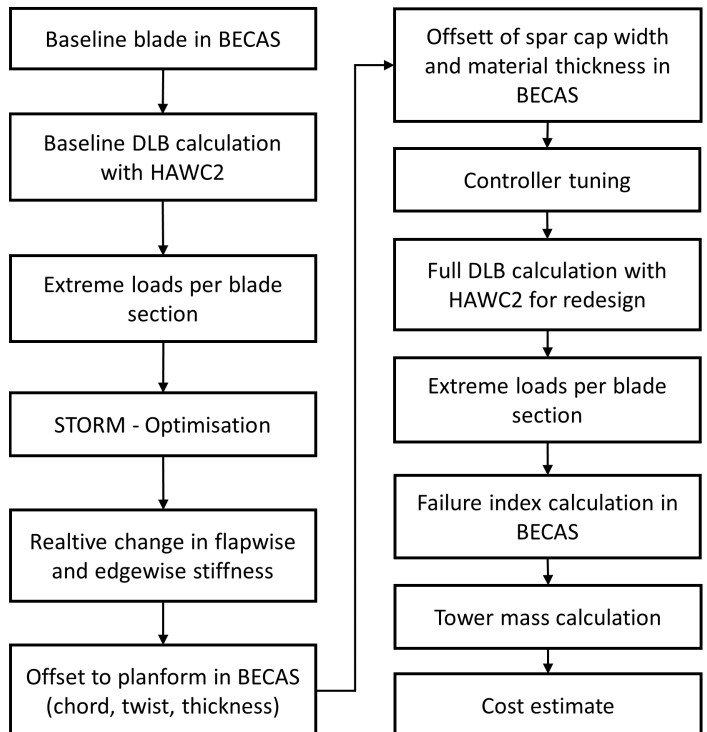

**Figure 1.** Flow chart of the work flow for design and cost estimation.

## 2.1 Baseline blade in BECAS

The baseline blade is set-up in BECAS (a 2D cross-sectional analysis tool, Blasques (2011)), as it is implemented in the HAW-

TOpt2 framework. This approach has several advantages. Firstly, through BECAS it allows having reasonable structural blade properties, which can be directly exported as inputs for load simulations with HAWC2. Secondly, even though the HAWTOpt2 framework is not utilized for optimization, it allows to parametrically modify the planform and structural dimensions of the baseline blade. Within the workflow of the framework the structural properties are recalculated within BECAS and can be exported for load calculations.

The baseline blade is described according to the parameterization adopted in HAWTOpt2 Zahle et al. (2016). To set-up the baseline blade in BECAS, the planform geometry is directly loaded into the framework. From blade length, relative thickness, chord, twist and the airfoil geometry the blade surface created as a 3D lofted surface. The chordwise position of the main structural regions, namely trailing edge caps, spar caps, shell panels, trailing edge and leading edge are defined in 19 cross-sections relative to the chord length. For each region, the positioning and material stacking is applied from blade lay-up plans.

For the baseline blade structural properties, total mass, static mass moment and blade eigenfrequencies are compared to the





commercial blade to assure the baseline is reasonable and fairly close to the commercial blade. The same has been done with turbine eigenfrequencies and damping, as well as the design driving loads for blades, main bearing, and tower.

## 2.2 Design load basis and controller definition.

Full design load bases are simulated with HAWC2 (version 12.7) according to the IEC-standard 61400-1 Edition 3 (IEC
(2014)). The interpretation of the design load basis by the Technical University of Denmark (DTU), described by Hansen et al. (2015), is used. For the downwind configuration the load simulations are conducted with an inflow inclination angle of 0°. The combination of positive flow inclination angle and turbine tilt was seen to be beneficial in downwind configurations by Wanke et al. (2019a). It is therefore assumed to be a more realistic scenario to simulate wind fields without inclination angle for downwind configurations.

The annual energy production (AEP) is calculated for all designs. It is calculated from the normal operation load case with six turbulence seeds, for all configurations without inclination or yaw angle. The turbulence intensity is according to the IEC-standard class A.

For all load calculations, the controller set-up from DTU (Hansen and Henriksen (2013)) is used in this study with two additional features. The controller is for pitch regulated variable speed turbines with partial and full load regions. Optimal $C_p$-
tracking is used in the partial load region and a constant torque strategy in the full load region. The detailed description of the controller can be found in Hansen and Henriksen (2013) and the source code is freely available online (https://github.com/DTUWindEnergy/BasicDTUController). The controller has been automatically tuned using a pole placement routine implemented in HAWCStab2 (Hansen (2004)), and which is described in more detail by Tibalidi et al. (2014). Different events can be initiated from the main controller, such as start-up and shut-down or failure situations.

Start-up and shut-down pitch speed in the implemented routines of the DTU-controller need different values for downwind configurations than comparable upwind configurations. The moment due to thrust force and the gravity overhanging moment of the rotor nacelle assembly both increase the tower bottom bending moment. Start-up routines, especially at high wind speeds, need to have a lower pitch speed in downwind configurations than the comparable upwind configurations. Shut-down routines, especially during gusts, have to be of faster pitch speed in the downwind configuration. Both adjustments have to be made to
unload the tower bottom.

For a control routine that reflects an industrial controller, three failure scenarios are adapted. Firstly, the failure scenario of one blade getting stuck at a current pitch angle, the pitch angle of one blade is kept constant at the current pitch angle at the time of failure. The deviation of the pitch angle from the setpoint initiates a stop routine of the turbine. Secondly, the pitch run away (DLC 2.2p) is not included, since the failure mode is prevented by the type of pitch actuators used. Thirdly, for the scenario of
a parked turbine with high yaw errors the wind field is interpreted as a wind direction change of 360° over 570 seconds.

To eliminate fault cases from the design driving loads, and to stay similar to an industrial controller, two additional control features are implemented as separate dlls manipulating the output or input from the controller to HAWC2, for practical reasons. The first addition is a thrust control aiming to reduce fluctuations of the thrust. The second addition is a conditional stop routine avoiding operation at high yaw errors and high wind speeds. The following explains the two additions in more detail.



Figure 2 shows the thrust control feature. The thrust control uses the sum of the flapwise blade root moments to estimate the

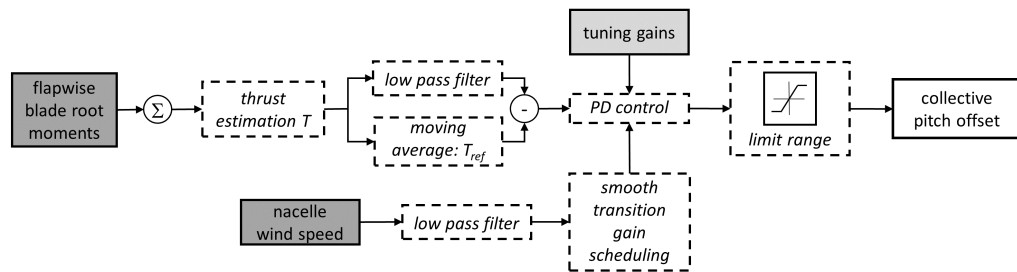

**Figure 2.** Flow chart of the thrust control controller addition.


thrust $T$. The thrust signal is on one hand low pass filtered and on the other hand, a reference thrust $T_{ref}$ is calculated as a moving average. A PD-controller is used to generate a collective pitch offset under range limitation. The pitch range is limited to avoid high loads from turbulence (lower limit) and high power losses (upper limit). The filtered wind speed of the nacelle anemometer is used for wind speed dependent gain scheduling to guarantee a smooth transition between active and not active

thrust control.

The conditional stopping routine triggers the turbine stop, as soon as the filtered wind speed and the filtered wind direction are above a certain threshold. For practical reasons of implementation, the emergency stop is triggered.

### 2.3    STORM - Optimization

The re-design of the rotor blades is performed using the in-house code STORM (Suzlon Turbine Optimization fRaMework).

The code is aimed at fast conceptual rotor design optimization studies, and couples steady aerodynamic AEP considerations, with a simplified blade structural estimation. The code, written in Matlab, is organized as a nested optimization problem. The outer optimization loop controls the blade geometrical planform, and minimizes the blade mass subject to: linear constraints on the geometrical design variables, non-linear constraints on minimum AEP, and feasibility of all the inner optimization problems (Eq. 1).

In this study, the blade geometry design variables are limited to four spline control points that set the thickness-over-chord (TOC) ratios in fixed points along the blade span. The geometry at the blade root is fixed up to the point of maximum chord for all configurations. For each iteration of the outer optimization loop, six steps are taken; they are described in the following sections, and briefly consists of:

1. The blade TOC spline is defined from the control points (the four design variables).

2. The blade geometrical planform is outlined in terms of chord, twist and thickness distribution. An inner optimization returns the chord distribution that minimizes the squared difference from a target axial induction distribution.



3. Steady operational loads and the power curve are retrieved from a standard steady BEM formulation. An inner optimization sets the pitch angle to maximize aerodynamic power, subject to limitation on maximum: power, thrust, aerodynamic flapwise bending moment, and angle of attack (for stall considerations).

4. The steady BEM loads are scaled to extreme loads to be used in the structural optimization.

5. The blade structural properties are determined solving a fast low-fidelity structural optimization problem. The blade structure is simplified to two symmetric glass fiber spar-caps joined by an ellipse, Fig. 3. The inner optimization sets the thickness and width of the spar cap, and the ellipse thicknesses to minimize the blade static mass moment, subject to constraints on: maximum strain, maximum deflection, maximum linear buckling index, and design variables range.

6. Finally, the outer loop optimization objective function is evaluated. The estimated blade mass is here taken as objective function, and minimum AEP output is enforced as a non-linear constraint.

### 2.3.1   Outer optimization loop

The outer optimization problem (Eq. 1) is solved using the Matlab Pattern Search method (G. Kolda (2014)). The algorithm is set up to perform a complete search and pooling around the current point.

$$
\begin{aligned}
&\underset{\boldsymbol{a},\boldsymbol{t},\boldsymbol{e},\boldsymbol{h} \in \mathbb{R}^N}{\text{minimize}} \quad m(\boldsymbol{a},\boldsymbol{t},\boldsymbol{e},\boldsymbol{h}) \\
&\text{subject to} \qquad AEP(\boldsymbol{h}) \geq AEP^{min}, \\
&\qquad\qquad\qquad \delta(\boldsymbol{a},\boldsymbol{t},\boldsymbol{e},\boldsymbol{h}) \leq \delta^{max}, \\
&\qquad\qquad\qquad \varepsilon_i(a_i,t_i,e_i,h_i) \leq \varepsilon^{max}, \quad i=1,\ldots,N, \\
&\qquad\qquad\qquad \eta_i(a_i,t_i,h_i) \leq \eta^{max}, \quad i=1,\ldots,N
\end{aligned}
\tag{1}
$$


where $m$ is the mass of the blade, depending on the variables of spar cap width $\boldsymbol{a}=[a_1,\ldots,a_N]$, spar cap height $\boldsymbol{t}=[t_1,\ldots,t_N]$, the shell thickness $\boldsymbol{e}=[e_1,\ldots,e_N]$ and the section height $\boldsymbol{h}=[h_1,\ldots,h_N]$ at each of the $N$ cross sections. The constraints are a minimum AEP, a maximum blade deflection $\delta$, a maximum strain $\varepsilon$, and a maximum buckling coefficient $\eta$. A list of all formula symbols can also be found in Table A1 in the appendix.

The design variables are here the four thickness-over-chord (TOC) control points ratios. Linear constraints on the design variables are enforced to ensure they maintain within reasonable ranges, and that monotonically decreasing values are selected from root to tip. The objective function consists for this problem of minimizing the estimated blade mass, subject to non-linear constraints to: reach a minimum AEP output (as derived from the BEM steady power curves) and ensure feasibility in all the inner optimization problems.





### 2.3.2   Blade geometrical planform

Once the iteration TOC control points are fixed, the TOC distribution along the blade span is outlined with a Piecewise Cubic Hermite Interpolating Polynomial. A wind speed in the below-rated variable speed range is chosen, and the target axial induction distribution for the blade at that wind speed is fixed as an input. Similarly, also the angles of attack at which the airfoils are expected to operate at that wind speed point are also fixed. Both the target induction and the target angles of attack of the airfoils could in principle also be driven by the outer optimization loop but are considered as fixed inputs in this study.

With the given input set (TOC, target induction, target AoAs) the blade geometry is then retrieved in terms of chord, twist angle, and thickness for each section along the blade span. The chord is retrieved by solving a set of independent minimization problems (Eq. 2), one for each section along the blade span. The optimization objective is to minimize the square error between the target axial induction for that section, and the current induction, function of chord, subject to a linear constraint on the minimum and maximum chord.

$$
\begin{aligned}
&\underset{c_i \in \mathbb{R}^N}{\text{minimize}} \quad (ind_{target\ i} - ind_i)^2 \\
&\text{subject to} \quad c^{min} < c_i < c^{max}
\end{aligned}
\tag{2}
$$

In the current iteration the axial induction is retrieved from a steady BEM formulation, following Ning's implementation (Ning (2014)), where the BEM convergence is solved by minimizing a residual function of the flow angle. Once the chord is fixed, the twist angle is simply set as the difference between the converged flow angle returned by the BEM, and the input angle of attack for that section (minus eventually a chosen constant reference pitch angle).

### 2.3.3   Steady loads and power curves

Given the blade geometrical definition as from the step above, the steady power and loads curves are then determined running a standard steady BEM formulation, Hansen (2008), sweeping wind speeds between cut-in and cut-out. From the steady power curve, the Annual Energy Production (AEP) is retrieved accounting for the chosen wind speed distribution.

The operational pitch angle at each wind speed is retrieved from a simple optimization loop, where the objective is to maximize the aerodynamic power output, subject to constraints on: maximum power (the aerodynamic rated power), maximum thrust force, maximum aerodynamic blade flapwise bending moment, and minimum "stall distance" (Eq. 3). The latter is



defined as a minimum margin in degrees between the steady BEM angle of attack and the point of maximum lift for the corresponding airfoil; the stall constrained is only enforced for the outer 40 % of the blade span.

$$
\begin{aligned}
& \underset{\beta \in \mathbb{R}}{\text{maximize}} \quad P(\beta) \\
& \text{subject to} \quad P < P^{max}, \\
& \qquad\qquad T < T^{max}, \\
& \qquad\qquad M_{flap} < M_{flap}^{max}, \\
& \qquad\qquad \alpha < (\alpha^{max} - \alpha_{stall\ distance})
\end{aligned}
\tag{3}
$$

where $\beta$ is the pitch angle, $P$ is the aerodynamic power, $T$ is the thrust force, $M_{flap}$ is the flapwise bending moment. The angle of attack is $\alpha$ and $\alpha_{stall\ distance}$ is the "stall distance". In the case of this study, the constraint of maximum thrust and maximum aerodynamic flapwise bending moment are not active.

### 2.3.4   Loads scaling

The maximum aerodynamic steady flapwise bending moment is retrieved from the step above, and is scaled up to an extreme load using a ratio retrieved from full DLB HAWC2 simulations of the baseline blade:

$$
M_{extreme} = M_{extreme\ baseline} \frac{M_{BEM}}{M_{BEM\ baseline}}
\tag{4}
$$

$M_{extreme\ baseline}$ is the extreme load distribution of the baseline rotor, extracted from full DLB simulations in HAWC2 for the baseline blade. The distribution is fitted with a fourth-order polynomial to ensure that it can be differentiated. The $M_{BEM\ baseline}$ moment is the corresponding maximum steady BEM model retrieved for the same baseline blade.

In the case of the downwind configuration, a second flapwise design load case for cut-out wind speed is considered, as the minimum tower-blade clearance arises in different loading conditions. The load distribution for the maximum deflection towards the tower $M_{extreme\ deflection}$ is thus scaled from the baseline loads at cut-out wind speed as:

$$
M_{extreme\ deflection} = M_{extreme\ baseline\ deflection} \left(2 - \frac{M_{BEM}}{M_{BEM\ baseline\ wsp\ out}}\right)
\tag{5}
$$

For the downwind configuration, a decrease in the loading results into a larger deflection towards the tower.

The edgewise loads remain unscaled, as they are driven by the aerodynamic torque as well as the gravity load.

### 2.3.5   Blade structural design

The simplified blade structural model is based on the work of Blasques and Stolpe (2012), also presented in the thesis work of Carstensen (2017) and Andersen (2018). The blade is described as a sequence of beam elements, each with a cross-section simplified to the elements shown in Fig. 3. The main load-carrying structure is simplified as a symmetric girder with two glass-reinforced-plastic (GRP) spar caps, connected by a GRP ellipse. The ellipse major axis is taken equal to the section chord, and





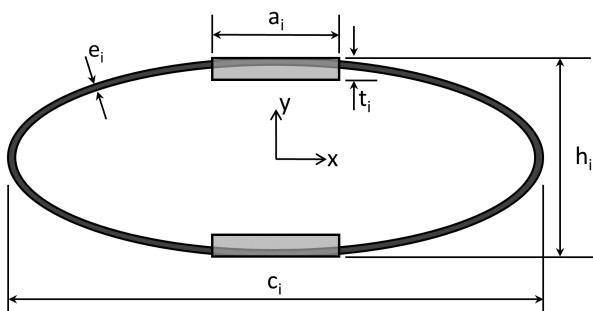

**Figure 3.** Simplified model of the blade structure for each cross section, as applied in STORM. The section height $h$ and chord length $c$ are fixed with the blade geometry for each iteration. The structural optimization design variables are then for each section: the spar caps thickness $t_i$ and width $a_i$, and the ellipse thickness $e_i$.

the distance between the spar caps taken equal to the section height, thus coupling the structural problem to the aerodynamic planform definition. The structural optimization problem has thus three design variables for each structural section $i$ along the blade: the spar cap thickness $t_i$ and width $a_i$, and the ellipse thickness $e_i$.

The load cases described in the previous section are applied to the finite beam element model, and the structural optimization aims at minimizing the blade static-mass moment, subject to constraints on: range of the design variables, maximum strain levels on caps and ellipses, maximum tip deflection for the deformed blade and maximum buckling coefficient for a single spar cap. The structural optimization problem can be stated as

$$
\begin{aligned}
&\underset{\boldsymbol{a},\boldsymbol{t},\boldsymbol{e},\in\mathbb{R}^N}{\text{minimize}} \quad m\left(\boldsymbol{a},\boldsymbol{t},\boldsymbol{e},\right) \\
&\text{subject to} \quad \delta\left(\boldsymbol{a},\boldsymbol{t},\boldsymbol{e},\right) \leq \delta^{max}, \\
&\qquad\qquad\quad \varepsilon_i\left(a_i,t_i,e_i,\right) \leq \varepsilon^{max}, \quad i=1,\ldots,N, \\
&\qquad\qquad\quad \eta_i\left(a_i,t_i,\right) \leq \eta^{max}, \quad i=1,\ldots,N
\end{aligned}
\tag{6}
$$

with the tip deflection for the deformed blade $\delta$, the strain levels on caps and ellipses $\varepsilon$ and the buckling coefficient $\eta$.

The buckling coefficient is added to the optimization problem compared to the references. The buckling coefficient is calculated under the assumption of an orthotropic plate under compression load. The compression load $N_z$ is obtained from the bending moment, assuming that the internal flapwise bending moment $M_x$ can be distributed as two forces acting on one girder side as compression forces and on the other girder side as tension forces.

$N_z = \dfrac{M_x}{h}\dfrac{1}{a}$         (7)



The buckling coefficient $\eta$ is then obtained via

$$\eta = \frac{6M_x}{\pi^2 Qh} \frac{a}{t^3} \tag{8}$$

$$Q = \frac{\nu_{12}E_2}{1 - \nu_{12}\nu_{21}} + 2G_{12} + \sqrt{\frac{E_1}{1 - \nu_{12}\nu_{21}} \frac{E_2}{1 - \nu_{12}\nu_{21}}}$$

with the elastic modulus $E$, the Poisson ration $\nu$, and the shear modulus $G$.

The optimization is solved with the Interior Point Optimizer Ipopt (Wächter and Biegler (2006)), and analytical gradients are given for objective and the constraints functions, thus speeding up considerably the process (Blasques and Stolpe (2012)). The solution returns a reliable estimate of the overall blade mass (and hence cost), which is here taken as the objective for the outer optimization loop.

## 2.4   Design evaluation

The optimized planform (chord, twist, and thickness distribution), as well as the changes in the structural geometry (spar cap width, thickness of the spar and trailing edge caps), are applied in HAWTOpt2 according to the planform calculated by STORM. All thickness distributions are fitted by hand at 5 control points and a spline fit is applied in between the control points. The HAWC2 inputs are extracted from HAWTOpt2 and a DLB is calculated for each redesign. From the DLB the maximum load at each blade cross-section is extracted. The failure index is calculated with BECAS for each cross-section.

The design is accepted if the failure index $i_f$ is $-1 < i_f < 1$. The failure index calculated by BECAS is not used in the design process.

The DLB calculation is further used to calculate the tower wall thickness $w$ for a tubular tower of given outer wall diameter $D$. The tower is divided into 50 cross-sections and the outer diameter, as well as the load distribution, are varied linearly between tower top and tower bottom. Within a for-loop, the wall thickness is increased until the stress $\sigma_{steel}$ reaches the allowed stress

of the tower steel material.

$$\sigma_{steel} = \frac{M\ SF}{W_b} \tag{9}$$

Where the bending moment $M$ is the bending moment of the cross section, $SF$ is the safety factor for steel material and the $W_b$ is the section modulus calculated as

$$W_b = \frac{\pi}{32} \frac{D^4 - (D - 2w)^4}{D} \tag{10}$$

The iteration is done twice, once for the extreme loads and the according stress limit for steel, and once for the lifetime equivalent load from the fatigue calculation and the fatigue stress limit for steel. From the two resulting wall thicknesses, the maximum thickness is picked for each cross-section. Constant masses for the tower interior are added and kept the same as for the baseline. The new tower mass distribution, as well as the stiffness redistribution, do not enter the DLB calculations.





## 2.5 Cost estimation

The cost model used for the cost evaluation consists of costs that scale with the mass, such as tower and blade costs. For other components, the costs scale with a design driving load or measure called cost driver $CD$. The cost driver is scaled with a factor $f_{CD\ to\ mass}$ to the component mass $Cost_{component}$. A second factor $f_{mass\ to\ cost}$ is defined to scale component masses to component costs.

$$Cost_{component} = f_{mass\ to\ cost}\left(f_{CD\ to\ mass}\ CD\right) \tag{11}$$

Other cost components e.g. logistics or operation and maintenance costs are scaled directly with the factor $f_{CD\ to\ mass}$ from a cost driver to the cost. Table 2 shows the cost drivers for the components entering the applied cost model. All component costs sum up to the capital expenditures (CAPEX). The operation and maintenance costs form the operational expenditures (OPEX). The OPEX costs are calculated with a net present value for a turbine lifetime of 20 years. The COE is calculated from the CAPEX, the OPEX and the AEP of 20 years lifetime.

$$COE = \frac{CAPEX + OPEX}{20AEP} \tag{12}$$

The component costs and total turbine costs (CAPEX+OPEX) of the baseline have been compared to the commercial turbine to assure a reasonable cost scaling and cost distribution within the present study.

## 3 Results

The following section presents the resulting design configurations regarding the planforms and resulting blade masses. Further,
the design driving loads and the resulting changes in turbine costs and COE are presented. All results are shown relative to the S111uw design configuration, as the data is confidential.

### 3.1 Design configurations

Figure 4 shows the planforms resulting from the design workflow. All values are normalized with the maximum chord. The figure shows that the chord and the twist distribution change only slightly, while larger differences can be observed for the thick-
ness over chord distribution, which is likely primarily due to the induction distribution being kept fixed during the optimization, while the larger changes in thickness are due to the direct coupling between AEP constraint, blade structural constraints and blade mass. For the S111uw PF and S111dw PF, the thickness over chord ratio increases from the 70% span and inboard compared to the baseline (S111uw). The S111dw PF has a slightly lower thickness than the S111uw PF design in this area up to the tip. From 40% span and inboard to the displayed region the S111dw PF design shows a larger thickness over chord ratio
than the S111uw PF design. In the outer 8% of the blade span, the PF redesigns show a larger thickness than the baseline blade. The latter is an artifact of the combination of the spline type chosen and the fixed airfoil thickness at the blade tip. For none of



**Table 2.** Cost drivers $CD$ for turbine cost and mass, split by main cost components (* indicates cost that are not scaled within the study due to CD)

| turbine component and cost design driver | |
| --- | --- |
| nacelle | |
| gear box incl. cooling* | nominal torque |
| pitch bearing | maximum static flapwise moment |
| main bearing | rotor static mass moment |
| main frame | extreme tilt moment |
| hub | extreme flapwise moment, blade static mass moment |
| main shaft | rotor own weight moment |
| gear rim | extreme yaw moment, tower top diameter |
| yaw drives | extreme yaw moment |
| pitch drives | maximum pitch moment, maximum pitch rate |
| converter* | nominal power |
| nacelle nose cone cover* | nominal power |
| power cable*s | nominal power, tower height |
| lift* | tower height |
| electrical | |
| generator* | nominal power |
| bottom panel* | nominal power |
| top panel* | nominal power |
| hub panel | maximum pitch moment, maximum pitch rate |
| transformer* | nominal power |
| blades | mass 70% (30% constant labor cost) |
| tower | mass |
| civil (foundation) | extreme tower bottom bending moment |
| cost component and cost design driver | |
| logistics | nacelle mass, blade length*, tower height* |
| electrical balance of plant | |
| yard* | blade length squared |
| electrical lines* | nominal power, average length of lines |
| installation (main crane) | nacelle mass times tower height |
| Operation and maintenance (OPEX)* | AEP |

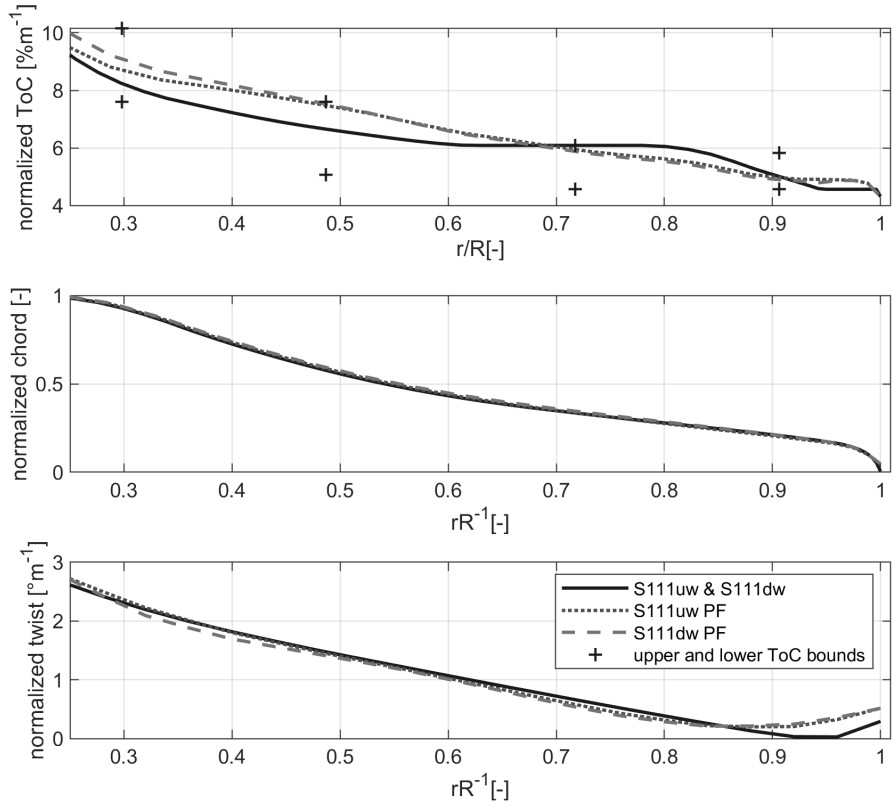

**Figure 4.** Comparison of planforms for different designs. Thickness over chord ratio as well as the range of the thickness constraints, chord and twist are normalized with the maximum chord.

the redesigns the constraint on thickness over chord is active.

While the S111uw PF design is constrained in blade deflection, in none of the downwind designs the blade deflection constraint is active. All the resulting redesigns are generally utilizing the maximum strain of the material over a larger blade span than
the S111uw and S111dw design configurations. All downwind redesigns are fully strain constrained in the spar caps. In the structural module of the optimization, the buckling constraint is active along the full blade span. The downwind configurations generally show larger shell thickness than the upwind configuration.

For all redesigns of the rotor blade, significant savings could be achieved of at least 12%. The lowest blade mass savings are achieved by the upwind configuration. For the S111uw PF 12.5% of blade mass could be saved. For the S111dw STR 14.5%
mass savings are achieved and 17.1% blade mass reduction was observed for S111dw PF. Table 4 summarizes the blade masses for all design configurations together with other achieved data.

The planform redesigns utilize higher stiffness with less material by using thicker airfoils in the inboard part. In the outboard





part thinner, more efficient airfoils compensate for a production loss of the inboard part of the blade. This effect is amplified as a small AEP penalty was allowed in the design procedure. From the S111dw STR, it can be seen that the downwind configu-

ration benefits from lower flapwise loads, and a release of the tower clearance constraint resulting in a reduced blade mass. A higher shell thickness is required to carry the higher edgewise loads in the downwind configurations. Comparing the S111dw PF design to the S111uw PF design a further effect of the edgewise load increase can be seen. To carry the increased edgewise loads there are two options. The first one is to increase the shell thickness as for the S111dw STR design. The second option is to increase the stiffness by using airfoils with higher relative thickness. The solution found in the optimization routine for

the S111dw PF is a combination of both, showing slightly thicker airfoils on the inboard part for the S111dw Pf than for the S111uw PF. Another solution to carrying the increased edgewise loads is an increased chord, but since the variation in chord is limited due to a fixed induction and tip speed ratio, this design freedom is not utilized. The lower flapwise loads in the S111uw PF design allow, on the other hand, to compensate for a power loss with slightly thinner airfoils in the outboard part. The chord distribution is hardly changing as the AEP is constrained to not deviate from the baseline AEP and the induction distribution is

frozen. The twist is simply adjusting the given operational point of the airfoils at the given spanwise position.

## 3.2 Cost driving loads from full DLB calculation

The following section shows the loads driving either the cost components in Table 2 or the designed tower and blade mass. For all regarded designs the minimum tower clearance is guaranteed. For all loads entering the cost model either directly or via the mass calculations DLC1.3, the operation at extreme turbulence remains design driving. The only exception is the extreme

blade root torsion moment where load cases of operation during wind direction change, operation at extreme yaw errors or yaw errors during parked situations with a locked rotor (DLC 1.4, DLC 2.2y or DLC 7.1) are design driving. Table 3 shows the loads influencing the cost estimation of the designs relative to the S111uw configuration. It can be seen, that the S111uw PF design clearly benefits, from the reduced blade mass on the edgewise extreme and fatigue blade root bending moment, as well as on the tower bottom bending moment. The only disadvantage is an increase in the tower top yaw moment.

The table also shows that the downwind designs generally benefit on the flapwise mean, flapwise extreme blade root moment and the related tower top yaw moment, mainly from the alignment of the rotor cone and the rotor forces ("load alignment"). In the tower top tilt moment, the influence of the tower shadow, as well as the alignment of the rotor overhanging gravity moment and the moment due to thrust force is observed. Due to the latter also an increase in the extreme tower bottom bending moment is seen compared to the S111uw design. The gravity-related loads, e.g. tower top tilt moment and longitudinal tower

bottom bending moment are reduced for each configuration by the reduction of mass due to the redesign. With the reduced flapwise stiffness of the S111dw PF design, the fatigue load is reduced to the level of the S111uw and the tower shadow effect is overcome. A relative reduction of the flapwise stiffness compared to the edgewise stiffness increases the edgewise damping which results in the load decrease for edgewise extreme and fatigue loads of the S111dw STR and S111dw PF compared to the S111dw.





**Table 3.** Turbine loads for mass and cost drivers, Blade root moment (BRM), tower bottom bending moment (TBM), tower top moment (TTM)

| load sensor | Δ normalized load relative to S111uw configuration | | | |
| --- | --- | --- | --- | --- |
| | S111uw PF | S111dw | S111dw STR | S111dw PF |
| max. mean flapwise BRM | -0.03 | -0.40 | -0.38 | -0.35 |
| extreme flapwise BRM | -0.01 | -0.17 | -0.19 | -0.18 |
| extreme edgewise BRM | -0.12 | +0.08 | +0.02 | -0.06 |
| extreme torsion BRM | -0.18 | +0.50 | -0.18 | -0.06 |
| extreme TTM yaw | +0.08 | -0.06 | -0.09 | -0.08 |
| extreme TTM tilt | -0.09 | +0.27 | +0.15 | +0.14 |
| extreme TBM longitudinal | -0.04 | +0.10 | +0.07 | +0.07 |
| fatigue flapwise BRM | -0.02 | +0.05 | +0.00 | +0.00 |
| fatigue edgewise BRM | -0.11 | +0.06 | +0.00 | -0.10 |
| fatigue TTM tilt | -0.01 | +0.06 | +0.06 | +0.05 |
| fatigue TBM longitudinal | -0.01 | -0.05 | -0.07 | -0.07 |

## 3.3 Turbine mass, cost and COE estimate

This section shows the estimated costs resulting from the load and mass difference of the design configurations. Figure 5 shows the summary of the main cost components of the turbine with an indication of the cost that is not affected by the design process (constant cost). The costs sum up to the total CAPEX. All results are normalized by the CAPEX of the S111uw design. It can be seen that the nacelle is the main cost component, followed by the blades, the tower, and the costs for electrical equipment. The figure shows that more than a third of the CAPEX is not affected by the chosen redesigns. In the CAPEX distribution of the nacelle, major cost differences are associated with the pitch bearing, the mainframe and the pitch drives. The blade costs reduce significantly with the redesign of the blades. Where the S111dw PF shows the lowest blade costs associated with the lowest blade mass. The tower and foundation costs are for the downwind configurations generally higher than for the upwind configurations, as the associated extreme loads and also the tower top fatigue loads are significantly higher. The costs of the electrical components reflect the change in hub panel costs as these scale with the extreme blade root torsion. Only small differences in the logistics costs are observed due to the change in nacelle mass. The balance of the plant is achieved for the same estimated costs while the installation reflects the changes in total main frame mass, driven by the extreme tilt moment. Overall, the total CAPEX costs of the turbine vary only marginally between all the redesigns. The OPEX costs, on the other hand, are lower for all the downwind designs since the OPEX costs scale with the lower AEP. As a result, the combined turbine



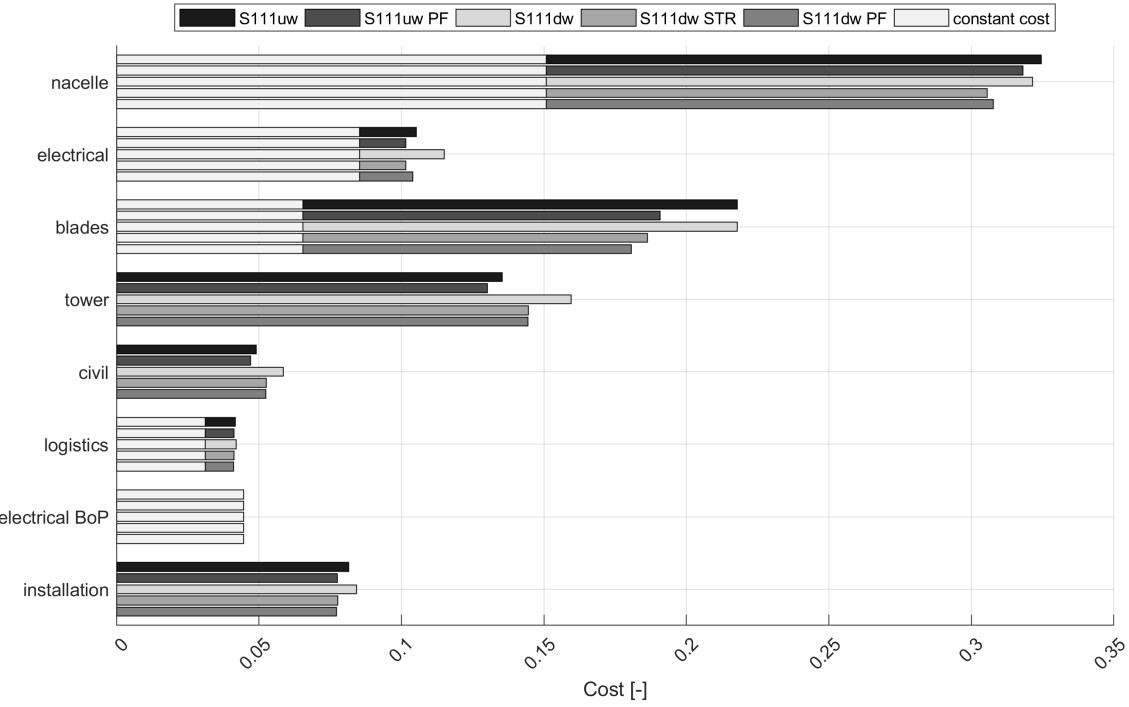

**Figure 5.** Turbine CAPEX cost split by main cost components normalized by the sum of the S111uw configuration with indication of constant costs not affected by redesign process.

costs of the redesigned downwind configurations (S111dw STR and S111dw PF) are lower than for the S111uw PF design.
Table 4 summarizes the achieved blade and tower mass, as well as the AEP and the estimated COE differences compared to the
S111uw design. With a COE reduction of -2.3% the S111uw PF shows the lowest COE, as the CAPEX is low, while the AEP

**Table 4.** Blade mass, Tower mass, CAPEX, AEP and COE difference for the regarded turbine configurations, relative to the S111uw design.

| Name | Δ blade mass [%] | Δ tower mass [%] | Δ AEP [%] | Δ COE [%] |
|---|---|---|---|---|
| S111uw PF | -12.5 | -3.9 | -0.33 | -2.3 |
| S111dw | 0.0 | +17.8 | -2.0 | +3.1 |
| S111dw STR | -14.5 | +6.8 | -2.32 | -1.2 |
| S111dw PF | -17.1 | +6.6 | -2.37 | -1.3 |

is high. A pure configuration change from S111uw to S111dw is most expensive in terms of COE, due to the high CAPEX
mainly caused by high tower and foundation loads. A structural redesign of the blades for the downwind configuration achieves
significant COE savings of -1.2%, due to reduced rotor mass. A planform optimization of the downwind configuration reduces





the COE -1.3% below the S111uw baseline turbine. Overall, the S111uw PF remains of lowest COE, since the rotor mass is only 5% above the S111dw PF while the tower is 10% lighter and the AEP is 2% higher.

## 4    Summary

Within this study, the COE reduction potential for the Suzlon S111 2.1MW turbine has been estimated changing the original

upwind configuration into a downwind configuration. A design framework including a low fidelity in-house optimization tool has been used to redesign rotors for upwind and downwind configurations. A full design load basis has been simulated for every design configuration. The design configurations have been evaluated by a COE estimation.

New planforms were optimized for upwind and downwind configurations for minimum blade mass under the constraint of a minimum AEP. The new planforms were shown to have higher thickness over chord ratios inboard, utilizing higher stiffness

with less material. This design trend agrees well with findings by Bottasso et al. (2016) and Zahle et al. (2016).

The downwind design were generally subject to lower flapwise blade root moments than the comparable upwind designs, due to the coning direction, as also proposed by for example Ichter et al. (2016) and Bortolotti et al. (2019). As a result lower blade mass could be achieved for downwind configurations than for upwind configurations. The S111dw PF design showed, for example, 4.6% lower blade mass than the S111uw PF design.

The load saving on the blade in the downwind configuration is offset by an increase in the tower bottom bending moment as the gravity overhanging moment of the rotor nacelle assembly is aligned with the thrust force, as also shown by Ning and Petch (2016). As a result around 10.5% higher tower masses were seen in the direct comparison of the S111uw PF design and the S111dw PF design.

The downwind configurations are subject to a lower AEP production due to the coning direction. This effect has also been

observed by for example Zalkind et al. (2019) or Ning and Petch (2016). In the direct comparison, the AEP of the S111dw PF is 2.04% lower than the AEP of the comparable S111uw PF.

Lower rotor and nacelle costs can be achieved by the downwind designs. However, the downwind designs also come with higher tower and foundation costs. Overall, the downwind configurations of comparable rotor size achieve a lower total turbine cost than the upwind design configuration. The difference in cost is due to the lower OPEX cost and does heavily depend on the

cost model. Overall the lower turbine cost does not compensate for the loss in AEP. The lowest COE level is achieved by the S111uw PF design configuration which achieves a significant mass and load reduction for a small sacrifice in AEP compared to the baseline.

## 5    Discussion and future work

This study has shown, for the example of the Suzlon S111 2.1MW turbine, that a downwind rotor configuration could be

achieved with lower total turbine costs than the comparable upwind configuration. Due to a lower AEP of the downwind configurations, the upwind configuration, on the other hand, showed overall the lowest COE. A downwind configuration would,





therefore, be the configuration to choose on a turbine cost-driven market, while for COE driven markets the upwind configuration would be chosen.

These results depend on the very baseline specific cost model. Scaling the OPEX with the AEP has been the only cost driver
for the OPEX which results in the lower turbine costs for the downwind configuration. It could be expected, that the higher fatigue load of the downwind configuration would increase the material wear, but this does not enter the OPEX model.

The cost model generally depends on the loads simulated. This comes with uncertainty due to the seed number, the seeds themselves as well as and the assumptions of wind field inclination angle. In the case of the downwind configuration additionally, the dynamic effect of the tower shadow is not captured correctly within the HAWC2 simulations. Within HAWC2 the tower
shadow model for downwind configurations is a pure deficit model and the increased vorticity behind the tower is not reflected. It can be expected that especially flapwise blade root and tilt related fatigue loads are under-predicted. Further research would need to be done to quantify the impact of this effect.

Generally fatigue loads should be part of the design process in future work. In the chosen approach fatigue loads are not regarded in the design process and hardly reflected in the cost model. This might be a valid assumption in the upwind con-
figurations, but for downwind configurations, this approach needs to be proven. Due to the tower shadow effect as well as a possible decrease in edgewise damping, it might be possible that rotors of downwind configurations are driven by edgewise fatigue loads rather than flapwise extreme loads.

Prescribing the induction distribution in the optimization is a major restriction of the chosen design approach. The resulting chord and twist distributions are therefore very similar. The induction should be a design variable in future work as unloading
the tip might allow for increases in rotor diameter and therefore AEP increase. The latter does not just hold for the downwind configuration, but also for the upwind configuration.

Prescribing the induction distribution did, however, have the advantage that the load scaling approach was possible. Scaling loads from the BEM-code loads to the extreme loads has decreased computation time significantly. A drawback of the load scaling approach is that a change in aerodynamic damping is not reflected. For the downwind configuration the flapwise stiff-
ness could be significantly reduced, while the edgewise stiffness had to be increased, the edgewise whirl modes can, therefore, be expected to increase in damping due to the frequency placement of the edgewise frequency compared to the second yaw frequency. An increase in damping decreases the blade extreme loads. The effect of the loads has been observed in the downwind designs, but there is no feedback within the optimization reflecting the change in damping. In future work, the framework would need to be enhanced with either time consuming load calculations or with a set of transfer functions that can transfer a
wind field to extreme loads from a linearized turbine model. Such a linearized turbine model could be extracted for example from HawcStab2 which uses these models for eigenvalue analysis. In this case, a representative wind field could be used that represents extreme loads from a simulation set with a much larger seed number and with known uncertainty. This would decrease the computational time drastically while achieving reasonable results.

The COE estimation and therefore success criteria of the downwind concept does also depend on the cost-share between the
different components. Since in the chosen example turbine the rotor and the tower are similar in the CAPEX share, it is difficult in the downwind configuration to offset the increased tower cost with savings on the rotor. If the baseline had a comparably





more expensive rotor and a cheaper tower, the downwind configuration would be more competitive. Possible scenarios could be lower steel prices or higher blade material prices.

Another possibility to increase the competitiveness of the downwind configuration would be a change in the tower config-
uration, such as a wired tower, where wires are a cheap measure to take the bending loads. Alternatively, a low labor cost market could give the options of low tower costs with lattice or hybrid-lattice towers which generate bending stiffness from an increased foot-print of the tower, rather than large tower wall thicknesses for a tubular tower. These options could make the downwind configuration competitive as the cost share of the tower decreases. However, the cost model with the chosen baseline is not able to reflect such drastic design changes.

Compensating the AEP loss in the downwind configuration with a larger rotor area could be an option to decrease the COE. Nevertheless, this does also increase the turbine cost, not just due to an increased rotor diameter and therefore rotor mass, but also mass related loads such as tilt loads and tower base loads. The rotor diameter has not been part of the rotor design as the cost model is very specific and does not reflect large differences from the baseline. Especially for components such as generator or gearbox which are not available in any possible configuration but are bought as "off-the-shelf" components the linear cost
scaling is insufficient. A rotor diameter increase of 4% has been investigated, indicating the potential to decrease the COE for the downwind configurations further with an increase of rotor diameter.

Future work should also consider a redesign of the nacelle for better balancing of the rotor mass on the tower for the downwind configuration, as suggested by Zalkind et al. (2019). However, it should be kept in mind, that the upwind configuration will always be beneficial in terms of tower bottom bending moment. Masses that can not be relocated for balancing such as rotor,
hub, pitch, and yaw system related masses account for around 50% of the mass of the rotor-nacelle assembly. Extending the lengthwise dimensions of the remaining components to relocate the center of gravity might be more expensive than the higher tower costs of downwind configurations.

It should not be forgotten in the discussion of the cost efficiency of downwind configurations, that simple control features such as peak shaving, as suggested by Loenbaek et al. (2019) might benefit the upwind configuration in the same manner as the
configuration change: the tower clearance is increased, the flapwise blade root moment is decreased with a penalty on AEP. Since the tower bottom load does in this case not increase as in the case of the downwind configuration, such an upwind configuration might out-perform a downwind configuration in terms of COE.

## 6 Conclusion

Overall, the study shows, that a downwind configuration of the chosen example 2.1MW turbine would need to be pushed to
much larger rotor sizes than investigated. Further, low-cost measures would need to be chosen to carry the increased tower loads if the downwind configuration should become competitive in terms of COE with the comparable upwind configuration. The optimization framework would need to be extended to be able to capture the design changes regarding the rotor, but also different tower configurations need to be included. To be able to evaluate such changes, a more comprehensive cost model is required to do a fair comparison of the designs.



It can be concluded from the study, that it will be difficult to design a downwind configuration in the 2MW range, which can show significant economic benefits unless the design targets a different market than the upwind configuration, or more drastic changes are made than just a rotor redesign and a structural redesign of the tower.

*Data availability.*   The data is not publicly accessible, since the research is based on a commercial turbine and the data is not available for disclosure by Suzlon.

**Appendix A:  List of Symbols**

The following Table A1 states the symbols used in the equations.

*Author contributions.*   GW has created the baseline model and implemented the downwind design case in the optimization framework. LB has developed and set-up the optimization framework. FZ has set-up HAWTOpt2 framework for baseline and design case evaluation. All authors have revised the models and results. With revisions of all co-authors GW and LB prepared section 2.3, GW prepared the remaining

paper.

*Competing interests.*   This project is an industrial PhD project funded by the Innovation Fund Denmark and Suzlons Blade Science Center. Gesine Wanke is employed at Suzlons Blade Science Center.





**Table A1.** List of symbols used in the equations

| Greek symbol | definition |
| --- | --- |
| $\alpha$ | angle of attack |
| $\beta$ | pitch angle |
| $\varepsilon$ | material strain |
| $\delta$ | blade deflection |
| $\eta$ | buckling coefficient |
| $\nu$ | Poisson ratio |
| $\sigma_{steel}$ | steel material stress |
| $\boldsymbol{\psi}$ | cross section strain and curvature vector |

| Latin symbol | definition |
| --- | --- |
| $a$ | spar cap width |
| $CD$ | cost driver |
| $D$ | outer tower diameter |
| $e$ | shell thickness |
| $E$ | elastic (Young's) modulus |
| $f$ | cost scaling factor |
| $G$ | shear modulus |
| $h$ | section height |
| $ind$ | induction |
| $m$ | mass |
| $M$ | local mending moment |
| $N$ | number of cross sections |
| $N_z$ | buckling load |
| $SF$ | safety factor |
| $t$ | spar cap thickness |
| $\boldsymbol{u}$ | displacement vector |
| $w$ | tower wall thickness |
| $W_b$ | section modulus |



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
