# Peer review of "Re-design of an upwind rotor for a downwind configuration: design changes and cost evaluation"

_Wind Energy Science, 2019_

## Referee Comment (RC1) · Pietro Bortolotti (Referee) · 2 Jun 2020

The paper runs a comparison between upwind and downwind configurations of a commercial 2MW wind turbine. Design optimization studies are run and the results are compared. The conclusions indicate an advantage of upwind in terms of COE and a possible minor advantage of downwind in CAPEX. The paper reads well and I recommend it for publication subject to minor revisions.

The design methods combine lower and higher fidelity approaches in a sequential approach that does not guarantee the true optimality of the design solutions. The choice of the objective functions is also confusing, with AEP in some steps treated as a con-

straint. It has been shown in https://doi.org/10.1115/1.4027693 that this can lead to suboptimal results. This, combined with the non dimensional presentation of the results, forces the reader to several acts of faith in your claims. This is a common problem of studies in the area of wind turbine design optimization, but I would prefer to see some disclaimers about this, highlighting a little better where some of the limitations of the present approach may be hiding. In my view, the biggest uncertainty is about rotor diameter and whether adding it among the design variables has the potential to reduce LCOE more for the downwind configuration than for the upwind. In lines 460-465 you let the reader understand that the outlook is pessimistic. However lines 465-466 say exactly the opposite. Please elaborate more about this aspect. Do you expect that your conclusions may change drastically? Do the downwind designs have some margins in the loads to support such increase in rotor diameter compared to the equivalent downwind? Table 3 suggests to me that no real margin is available. Another aspect where I'm not totally convinced consists of the analysis of the design drivers. At line 325 we read that the spar caps are strain constrained. Why doesn't then the optimizer reduce relative (and therefore absolute) thickness? If I understand right, Figure 4 shows the same absolute thickness between S111uw PF and S111dwPF. What are the design drivers at convergence? And if tip deflection is not an active design driver, why not reducing cone and tilt angles for the downwind? This would certainly help reducing the losses in AEP. Overall, a cleaner discussion on the different design drivers between upwind and downwind is needed in my opinion. Load analysis is great, but it's often somewhat nebulous.

Some smaller suggestions for improvements are listed below: 1) I suggest to reformulate the logical connection on lines 23-25 "Upwind rotor configurations, on the other hand, were found to be significantly less noisy. Upwind rotor configurations therefore dominated industrial applications as well as the focus in research efforts during the 1990s and 2000s.". As it is posed, it sounds like that it was only a noise problem. I think that the overall combination of the problems that you listed between lines 1-23 contributed to upwind dominating over downwind. 2) I suggest to reformulate

the sentence "Since the blade tip deflection constraint could be eliminated in modern sized wind turbines, the downwind configuration is currently coming into research focus again, especially for future even larger rotors.". The tip deflection constraint cannot be eliminated in modern wind turbines, even in downwind ones... We all know this, but I still find the sentence potentially confusing. 3) Line 80: define cp, which later becomes Cp 4) Line 84: check the sentence "This paper evaluates the specific example of the Suzlon S111 2.1MW turbine the potential of a downwind turbine configuration compared to the original upwind turbine configuration regarding mass and cost reduction." 5) Please improve the introduction by better defining the motivation behind this work. What are the gaps existing in literature? How do you plan to address them? How do you structure the analysis? Currently, the introduction consists of a long list of little summaries of relevant work, without however highlighting what is missing and how the paper contributes in terms of novelty. Why is this work important? The introduction should answer this question leaving no doubt to the reader. Also, lines 84-89 describe the conclusions. These should be in the abstract and in the conclusions, not in the introduction. 6) Line 131: "For the downwind configuration the load simulations are conducted with an inflow inclination angle of 0.". What about the upwind? Why not keeping the same angle as prescribed by the standards? If it gives an unfair advantage to downwind, why not sticking to 0 deg for both designs? Does it actually give an unfair advantage to downwind? 7) Line 147-149: "Start-up routines, especially at high wind speeds, need to have a lower pitch speed in downwind configurations than the comparable upwind configurations. Shut-down routines, especially during gusts, have to be of faster pitch speed in the downwind configuration." . Please explain these effects and how you address them in more detail. What are the limitations of the downwind configurations? How did you solve them? 8) Line 326: where do the highest blade deflection of the downwind designs occur? Is shutdown a problem as pointed out in several works? What about high wind speeds? And start ups? Please discuss the design drivers of both upwind and downwind better, so that the reader can gain confidence in these results. 9) Line 328: please revise the sentence "For all redesigns of the

rotor blade, significant savings could be achieved of at least 12%." Savings in what? 10) For some reason all figures were generated b/w. Some color would be great... 11) Line 364: I would move Table 3 to percentage values. Right now it is somewhat hard to interpret. Plots could also help. Same for Figure 5, % values would be more immediate to interpret. And please add some colors.

---

## Referee Comment (RC2) · Andrew Ning (Referee) · 20 Jul 2020

general comments:

- This paper investigates redesigning a specific turbine configuration from upwind to downwind. The writing is clear and the scope is appropriate. The paper is interesting, there are some contributions though the novelty is not high. The motivation and results are well explained. My main concerns are all with methodology. It is publication worthy, but need some additional work.

specific comments:

[Figure]

- The literature review is well done. Thorough summary/discussion of previous work. What is lacking is more motivation on why this paper. What is the gap you are addressing? What are the novel and impactful contributions? Not saying those don't exist, they just aren't discussed. All that is said is basically we are going to look at redesigning a specific turbine for downwind operation.

- The physics methods are all reasonable and explained. My main concerns are with the optimization methodology. Some parts were less clear so some may be my own misunderstanding, but these are the concerns:

1) the main process is some combination of sequential and/or nested optimization (not entirely clear the interaction between the different pieces). Both have well known convergence issues and can lead to non-optimal solutions - these are not discussed.

2) Related - convergence quality in general is not discussed. What tolerances were used for each of the optimizations (including sub problems)? With so many levels and sequential processes it does not seem likely that well-converged optimal points are found. That's fine for improvement - especially for an industrial application no one really cares if you've found "the optimum", but it is a bit more problematic if you are making claims on which design is better when the cost differences are small and likely within the margin of error. This can partially be addressed with just caveats on how to interpret the results.

3) The paper explained that the top-level objective was mass. That's fine, but then conclusions based on cost are likely misleading. It's ok to compare those, but you can't really draw conclusions on relative costs. In other words, if cost was thee top-level objective you would likely find a different design than one optimized for mass.

4) The design variables and the relationships between thee sequential/nested optimizations are not clear. How do you optimize with steady loads but also incorporate dynamic load cases? What are the various optimizations connected (inputs/outputs/feedback)? A formal diagram would really help.

Interactive
comment

5) What is the target axial distribution? Why is there a target and how was this decided?

6) How well does this loads scaling procedure work as the geometry changes? There should be some verification recomputing the moments directly with HAWC2 to show that the scaling approach is reasonable. Derivatives are mentioned here, what type of optimization is used? I don't think that is stated for Eq 3.

- The cost model is rather vague. There is a big list of variables in Table 2 but the reader has no idea how these are used. Reproducibility is not possible. Perhaps this is method is private, but at least some more detail would be helpful.

- Good discussion at end of paper.

technical corrections:

- Fig 5 would really benefit from color. hard to read.

- There are quite a few places with awkward phrasing or incorrect grammar. Overall it reads really well, but some additional proofing would help clean up some of these sentences that are hard to parse.

---

## Author Comment (AC1) · 15 Aug 2020

Dear Referees,

Thank you for your comments, they are very much appreciated. Please find the original comments as well as the answers and the manuscript with marked changes in the PDF attached.

Best regards, Gesine

Please also note the supplement to this comment:

https://wes.copernicus.org/preprints/wes-2019-102/wes-2019-102-AC1-supplement.pdf

---

## Author Response (AR1)

The paper runs a comparison between upwind and downwind configurations of a commercial 2MW wind turbine. Design optimization studies are run and the results are compared. The conclusions indicate an advantage of upwind in terms of COE and a possible minor advantage of downwind in CAPEX. The paper reads well and I recommend it for publication subject to minor revisions.

The design methods combine lower and higher fidelity approaches in a sequential approach that does not guarantee the true optimality of the design solutions. The choice of the objective functions is also confusing, with AEP in some steps treated as a con-

straint. It has been shown in https://doi.org/10.1115/1.4027693 that this can lead to suboptimal results. This, combined with the non dimensional presentation of the results, forces the reader to several acts of faith in your claims. This is a common problem of studies in the area of wind turbine design optimization, but I would prefer to see some disclaimers about this, highlighting a little better where some of the limitations of the present approach may be hiding. In my view, the biggest uncertainty is about rotor diameter and whether adding it among the design variables has the potential to reduce LCOE more for the downwind configuration than for the upwind. In lines 460-465 you let the reader understand that the outlook is pessimistic. However lines 465-466 say exactly the opposite. Please elaborate more about this aspect. Do you expect that your conclusions may change drastically? Do the downwind designs have some margins in the loads to support such increase in rotor diameter compared to the equivalent downwind? Table 3 suggests to me that no real margin is available. Another aspect where I'm not totally convinced consists of the analysis of the design drivers. At line 325 we read that the spar caps are strain constrained. Why doesn't then the optimizer reduce relative (and therefore absolute) thickness? If I understand right, Figure 4 shows the same absolute thickness between S111uw PF and S111dwPF. What are the design drivers at convergence? And if tip deflection is not an active design driver, why not reducing cone and tilt angles for the downwind? This would certainly help reducing the losses in AEP. Overall, a cleaner discussion on the different design drivers between upwind and downwind is needed in my opinion. Load analysis is great, but it's often somewhat nebulous.

Some smaller suggestions for improvements are listed below: 1) I suggest to reformulate the logical connection on lines 23-25 "Upwind rotor configurations, on the other hand, were found to be significantly less noisy. Upwind rotor configurations therefore dominated industrial applications as well as the focus in research efforts during the 1990s and 2000s.". As it is posed, it sounds like that it was only a noise problem. I think that the overall combination of the problems that you listed between lines 1-23 contributed to upwind dominating over downwind. 2) I suggest to reformulate

the sentence "Since the blade tip deflection constraint could be eliminated in modern sized wind turbines, the downwind configuration is currently coming into research focus again, especially for future even larger rotors.". The tip deflection constraint cannot be eliminated in modern wind turbines, even in downwind ones... We all know this, but I still find the sentence potentially confusing. 3) Line 80: define cp, which later becomes Cp 4) Line 84: check the sentence "This paper evaluates the specific example of the Suzlon S111 2.1MW turbine the potential of a downwind turbine configuration compared to the original upwind turbine configuration regarding mass and cost reduction." 5) Please improve the introduction by better defining the motivation behind this work. What are the gaps existing in literature? How do you plan to address them? How do you structure the analysis? Currently, the introduction consists of a long list of little summaries of relevant work, without however highlighting what is missing and how the paper contributes in terms of novelty. Why is this work important? The introduction should answer this question leaving no doubt to the reader. Also, lines 84-89 describe the conclusions. These should be in the abstract and in the conclusions, not in the introduction. 6) Line 131: "For the downwind configuration the load simulations are conducted with an inflow inclination angle of 0.". What about the upwind? Why not keeping the same angle as prescribed by the standards? If it gives an unfair advantage to downwind, why not sticking to 0 deg for both designs? Does it actually give an unfair advantage to downwind? 7) Line 147-149: "Start-up routines, especially at high wind speeds, need to have a lower pitch speed in downwind configurations than the comparable upwind configurations. Shut-down routines, especially during gusts, have to be of faster pitch speed in the downwind configuration." . Please explain these effects and how you address them in more detail. What are the limitations of the downwind configurations? How did you solve them? 8) Line 326: where do the highest blade deflection of the downwind designs occur? Is shutdown a problem as pointed out in several works? What about high wind speeds? And start ups? Please discuss the design drivers of both upwind and downwind better, so that the reader can gain confidence in these results. 9) Line 328: please revise the sentence "For all redesigns of the

rotor blade, significant savings could be achieved of at least 12%." Savings in what? 10) For some reason all figures were generated b/w. Some color would be great... 11) Line 364: I would move Table 3 to percentage values. Right now it is somewhat hard to interpret. Plots could also help. Same for Figure 5, % values would be more immediate to interpret. And please add some colors.

———————————————————

[Figure]

Wind Energ. Sci. Discuss.,
https://doi.org/10.5194/wes-2019-102-RC2, 2020

[Figure]

- This paper investigates redesigning a specific turbine configuration from upwind to downwind. The writing is clear and the scope is appropriate. The paper is interesting, there are some contributions though the novelty is not high. The motivation and results are well explained. My main concerns are all with methodology. It is publication worthy, but need some additional work.

specific comments:

[Figure]

- The literature review is well done. Thorough summary/discussion of previous work. What is lacking is more motivation on why this paper. What is the gap you are addressing? What are the novel and impactful contributions? Not saying those don't exist, they just aren't discussed. All that is said is basically we are going to look at redesigning a specific turbine for downwind operation.

- The physics methods are all reasonable and explained. My main concerns are with the optimization methodology. Some parts were less clear so some may be my own misunderstanding, but these are the concerns:

1) the main process is some combination of sequential and/or nested optimization (not entirely clear the interaction between the different pieces). Both have well known convergence issues and can lead to non-optimal solutions - these are not discussed.

2) Related - convergence quality in general is not discussed. What tolerances were used for each of the optimizations (including sub problems)? With so many levels and sequential processes it does not seem likely that well-converged optimal points are found. That's fine for improvement - especially for an industrial application no one really cares if you've found "the optimum", but it is a bit more problematic if you are making claims on which design is better when the cost differences are small and likely within the margin of error. This can partially be addressed with just caveats on how to interpret the results.

3) The paper explained that the top-level objective was mass. That's fine, but then conclusions based on cost are likely misleading. It's ok to compare those, but you can't really draw conclusions on relative costs. In other words, if cost was thee top-level objective you would likely find a different design than one optimized for mass.

4) The design variables and the relationships between thee sequential/nested optimizations are not clear. How do you optimize with steady loads but also incorporate dynamic load cases? What are the various optimizations connected (inputs/outputs/feedback)? A formal diagram would really help.

[Figure]

5) What is the target axial distribution? Why is there a target and how was this decided?

6) How well does this loads scaling procedure work as the geometry changes? There should be some verification recomputing the moments directly with HAWC2 to show that the scaling approach is reasonable. Derivatives are mentioned here, what type of optimization is used? I don't think that is stated for Eq 3.

- The cost model is rather vague. There is a big list of variables in Table 2 but the reader has no idea how these are used. Reproducibility is not possible. Perhaps this is method is private, but at least some more detail would be helpful.

- Good discussion at end of paper.

technical corrections:

- Fig 5 would really benefit from color. hard to read.

- There are quite a few places with awkward phrasing or incorrect grammar. Overall it reads really well, but some additional proofing would help clean up some of these sentences that are hard to parse.
* * *
[Figure]

Answer to Referee comments

**RC01 – Pietro Bortolotti**

Dear Pietro Bortolotti,

Thank you for your comments. They enhance our work and are very much appreciated. Due to the financial situation Suzlon has closed the Blade Science Center and we do no longer have access to the original data. We are therefore not able to modify the plots regarding color or normalization of values.

**Major comments:**

**The design methods combine lower and higher fidelity approaches in a sequential approach that does not guarantee the true optimality of the design solutions. The choice of the objective functions is also confusing, with AEP in some steps treated as a constraint. It has been shown in https://doi.org/10.1115/1.4027693 that this can lead to suboptimal results.**
*The only optimization is utilized as nested optimization with the low-fidelity tool STORM. The overall objective is therefore solved for in the outer loop, namely the minimization of blade mass under AEP constraint.*
*The high-fidelity tools are not used for sequential optimization but for design evaluation.*
*This is pointed out more clearly in the general description of the approach and the optimization.*

**Non dimensional representation of results.**
*As the baseline blade is a commercial blade, dimensional representations are not open for disclosure by Suzlon.*

**Highlighting more limitations of the chosen optimization approach**
*A paragraph is added in the discussion regarding the mass optimization under AEP constraint, the rotor diameter, and the limitations due to the prescribed design-induction approach.*

**In my view, the biggest uncertainty is about rotor diameter and whether adding it among the design variables has the potential to reduce LCOE more for the downwind configuration than for the upwind. In lines 460-465 you let the reader understand that the outlook is pessimistic. However, lines 465-466 say exactly the opposite. Please elaborate more about this aspect. Do you expect that your conclusions may change drastically? Do the downwind designs have some margins in the loads to support such increase in rotor diameter compared to the equivalent downwind? Table 3 suggests to me that no real margin is available.**
*The cost is lower than the S111dwPF design but not nearly as low as for the respective upwind design. The reason being that the rotor diameter increase comes not just with higher AEP but also with higher rotor and tower mass as well as higher nacelle costs due to the load increase. We do therefore not expect drastic changes in our conclusion.*
*The COE value for the additionally investigated rotor diameter and the conclusion is added.*

**Another aspect where I'm not totally convinced consists of the analysis of the design drivers. At line 325 we read that the spar caps are strain constrained. Why doesn't then the optimizer reduce relative (and therefore absolute) thickness?**

Because the lower mass will be achieved with larger thickness as more stiffness can be utilized with less material, without being penalized on the AEP side. If the section height was decreased the stress would increase and result therefore into a strain increase. Blade mass is the overall optimization objective. The latter is pointed out more clearly in the description of the optimization.

**If I understand right, Figure 4 shows the same absolute thickness between S111uw PF and S111dwPF.**

The absolute thickness is not the same. The chord is the same. As the AEP is constraint to the baseline AEP the optimization results into similar lift levels and as the induction distribution is frozen along the span, the resulting chord is the same. More explanation is added in the text.

**What are the design drivers at convergence? And if tip deflection is not an active design driver, why not reducing cone and tilt angles for the downwind? This would certainly help reducing the losses in AEP. Overall, a cleaner discussion on the different design drivers between upwind and downwind is needed in my opinion.**

The only difference in the design drivers at convergence is that the upwind designs are tip deflection constraint and therefore only over some part of the blade span strain constraint, while the downwind configurations are fully strain constraint. This is due to the fact, that the design driving situations are at different operational points. The remaining tower clearance is however only marginally for the S111dwPF design. The decrease in tilt or cone angle for larger AEP capture were not considered as variables. Due to the very small margin in tower clearance no significant gain in AEP can be expected.

A paragraph for a comparison for the active design drivers between S111uwPF and S111dwPF is added.

**Load analysis is great, but it's often somewhat nebulous.**

The section is rewritten for clarification.

**Minor comments:**

**1) I suggest to reformulate the logical connection on lines 23-25**

Logical connection is reformulated

**2) I suggest to reformulate the sentence "Since the blade tip deflection constraint could be eliminated in modern sized wind turbines, the downwind configuration is currently coming into research focus again, especially for future even larger rotors."**

The sentence is reformulated

**3) Line 80: define cp, which later becomes Cp**

Cp is defined

**4) Line 84: check the sentence "This paper evaluates the specific example of the Suzlon S111 2.1MW turbine the potential of a downwind turbine configuration compared to the original upwind turbine configuration regarding mass and cost reduction."**

Paragraph is reformulated.

**5) Please improve the introduction by better defining the motivation behind this work. What are the gaps existing in literature? How do you plan to address them? How do you structure the**

**analysis? Currently, the introduction consists of a long list of little summaries of relevant work, without however highlighting what is missing and how the paper contributes in terms of novelty. Why is this work important? The introduction should answer this question leaving no doubt to the reader. Also, lines 84-89 describe the conclusions. These should be in the abstract and in the conclusions, not in the introduction.**

Paragraph is reformulated to state motivation and novelty of the work.

**6) Line 131: "For the downwind configuration the load simulations are conducted with an inflow inclination angle of 0.". What about the upwind? Why not keeping the same angle as prescribed by the standards? If it gives an unfair advantage to downwind, why not sticking to 0 deg for both designs? Does it actually give an unfair advantage to downwind?**

Using the inclination angle of 8° as according to the standard would be an advantage for the downwind configuration as the tilt angle would align for a more even inflow on the rotor plane. An inclination angle of 0° has been checked to give representative turbine loads, without a major advantage on the load for the downwind configuration. The upwind configuration is still subject to the inclination angle of 8° as required by the standard.

Explanations are added to the text

**7) Line 147-149: "Start-up routines, especially at high wind speeds, need to have a lower pitch speed in downwind configurations than the comparable upwind configurations. Shut-down routines, especially during gusts, have to be of faster pitch speed in the downwind configuration." . Please explain these effects and how you address them in more detail. What are the limitations of the downwind configurations? How did you solve them?**

The reason is that the moment due to the thrust force is aligned with the overhanging rotor moment due to gravity. Therefore, pitching fast at gust situations reduces the tower base loads. On the other hand, during start-up a slower pitch velocity avoids a thrust overshoot and related high tower loads.

Explanations are added in the text.

**8) Line 326: where do the highest blade deflection of the downwind designs occur? Is shutdown a problem as pointed out in several works? What about high wind speeds? And start ups? Please discuss the design drivers of both upwind and downwind better, so that the reader can gain confidence in these results.**

Adjusting the pitch speed for downwind rotor configurations assures that the highest tip deflection does not occur in start-up or shut-down situations for this turbine (class IIIA). The operation at high wind speeds and high turbulence level (DLC1.3) is design driving for the minimum tip to tower distance in the downwind configuration, as the outboard part of the blade is subject to negative lift forces. High pitch angles at high wind speeds cause the negative lift force and the blade tip bends towards the tower. For the upwind configuration, the operation at the thrust peak at high turbulence (DLC1.3) is design driving for the minimum tip to tower distance.

More explanations are added to the load description explaining the design driving situations.

**9) Line 328: please revise the sentence "For all redesigns of the C3rotor blade, significant savings could be achieved of at least 12%." Savings in what?**

Sentence is changed.

**10) For some reason all figures were generated b/w. Some color would be great. . .**

Due to the situation at the Blade Science Center we do no longer have access to the data to change the plots. (The original reason for b/w figures was that 8% of the male human population are color blind).

**11) Line 364: I would move Table 3 to percentage values. Right now it is somewhat hard to interpret. Plots could also help. Same for Figure 5, % values would be more immediate to interpret. And please add some colors.**

Table 3 has been moved to percentage values. However, as we have no more access to the data plots cannot be changed.

Dear Andrew Ning,

Thank you for commenting our work. Your comments are very much appreciated and enhance our work. Due to the financial situation of Suzlon the Blade Science Center has been closed and we do no longer have access to the original data to change for example plots.

**Specific comments:**

**What is lacking is more motivation on why this paper. What is the gap you are addressing? What are the novel and impactful contributions?**
The main motivation is to investigate design trend differences between the upwind and downwind configuration and their impact on the design loads and turbine cost. This allows for a discussion on the potential of a downwind rotor configuration for the example turbine.
A paragraph is added to state the motivation more clearly.

**The physics methods are all reasonable and explained. My main concerns are with the optimization methodology. Some parts were less clear so some may be my own misunderstanding, but these are the concerns:**
Generally the only optimization is the nested optimization for blade mass within the STORM tool. Everything else is a post processing of the resulting structural properties. This has been pointed out for clarification

**1) the main process is some combination of sequential and/or nested optimization (not entirely clear the interaction between the different pieces). Both have well known convergence issues and can lead to non-optimal solutions - these are not discussed.**
The only optimization is the nested optimization in the low fidelity tool STORM. It has been checked that the optimization algorithm has converged within the given tolerances.  This guarantees only that the optimum within each step of the loop is found not necessarily that overall a global optimum is found. However, using the same approach for both configurations allows to investigate the design trends of the two configurations.
This is clarified in the text and added together with the flow diagram (see point 4) and also added to the discussion

**2) Related - convergence quality in general is not discussed. What tolerances were used for each of the optimizations (including sub problems)? With so many levels and sequential processes it does not seem likely that well-converged optimal points are found. That's fine for improvement - especially for an industrial application no one really cares if you've found "the optimum", but it is a bit more problematic if you are making claims on which design is better when the cost differences are small and likely within the margin of error. This can partially be addressed with just caveats on how to interpret the results.**
This is a fair point, especially since the results from the optimization are postprocessed for the load assessment. We have no more access to the inputs and are not able to give the numbers for the tolerances, but the same tolerances were of course used for both configurations.

**3) The paper explained that the top-level objective was mass. That's fine, but then conclusions based on cost are likely misleading. It's ok to compare those, but you can't really draw conclusions on relative costs. In other words, if cost was thee top-level objective you would likely find a different design than one optimized for mass.**

We do agree with the fact that the design objective change would change the design and the conclusion on the cost. But we can see from the results, that there is a trend that the upwind design would have a higher potential in cost saving than the downwind design, if the rotor diameter is not part of the optimization. As major cost contributions are due to the tower, nacelle, and the AEP loss due to coning, the trends should be captured. It is highlighted in the discussion that the values showing trends rather than absolute numbers.

**4) The design variables and the relationships between thee sequential/nested optimizations are not clear. How do you optimize with steady loads but also incorporate dynamic load cases? What are the various optimizations connected (inputs/outputs/feedback)? A formal diagram would really help**

A diagram and the description are added.

**5) What is the target axial distribution? Why is there a target and how was this decided?**

The target axial induction is kept according to the original commercial blade. It has been assumed that this is a typical induction distribution resulting from a commercial aerodynamic design process. It has been kept as the optimization tool is not capable of reflecting the complexity of fully variable induction regarding also related concerns such as stability or stall margins. This has been added in the methods.

**6) How well does this loads scaling procedure work as the geometry changes? There should be some verification recomputing the moments directly with HAWC2 to show that the scaling approach is reasonable. Derivatives are mentioned here, what type of optimization is used? I don't think that is stated for Eq 3.**

In inner optimization loop is a gradient based optimization with analytical gradients. The loads are fitted with a polynomial to assure differentiability. This is used to derive section forces from the bending moments. The loads are simply calculated with Equ. 3 thus, scaling the polynomial with a constant factor. The approach has only been checked by checking that the resulting tower clearance is captured. The failure indices calculated with BECAS also indicate that the sectional loads scaling approach works. The load scaling approach works fairly well, because only operational loads are considered in the optimization and the induction is prescribed, leading to a scalable load distribution. This is added in the text for clarification.

**- The cost model is rather vague. There is a big list of variables in Table 2 but the reader has no idea how these are used. Reproducibility is not possible. Perhaps this is method is private, but at least some more detail would be helpful.**

All the variables are used in linear equations to calculate a cost distribution (Equ. 11). The coefficients are tuned for the Suzlon turbines and are not available for disclosure (and also no longer available to us).

**- Good discussion at end of paper. technical corrections: - Fig 5 would really benefit from color. hard to read.**

Due to the situation of the blade science center we have no longer access to the data and are unfortunately not able to change the figures.

**- There are quite a few places with awkward phrasing or incorrect grammar. Overall it reads really well, but some additional proofing would help clean up some of these sentences that are hard to parse.**

The phrasing has been checked again.

[revised manuscript text omitted]